# Ambient Monitoring Portable Sensor Node for Robot-Based Applications

**DOI:** 10.3390/s24041295

**Published:** 2024-02-17

**Authors:** Mohammed Faeik Ruzaij Al-Okby, Steffen Junginger, Thomas Roddelkopf, Jiahao Huang, Kerstin Thurow

**Affiliations:** 1Center for Life Science Automation (Celisca), University of Rostock, 18119 Rostock, Germany; mohammed.al-okby@celisca.de (M.F.R.A.-O.); jiahao.huang@celisca.de (J.H.); kerstin.thurow@celisca.de (K.T.); 2Technical Institute of Babylon, Al-Furat Al-Awsat Technical University (ATU), Kufa 54003, Iraq; 3Institute of Automation, University of Rostock, 18119 Rostock, Germany; thomas.roddelkopf@uni-rostock.de

**Keywords:** sensor node, mobile robots, hazardous gases, harmful gases, volatile organic compounds (VOCs), ambient monitoring, environmental monitoring, gas sensors

## Abstract

The leakage of gases and chemical vapors is a common accident in laboratory processes that requires a rapid response to avoid harmful effects if humans and instruments are exposed to this leakage. In this paper, the performance of a portable sensor node designed for integration with mobile and stationary robots used to transport chemical samples in automated laboratories was tested and evaluated. The sensor node has four main layers for executing several functions, such as power management, control and data preprocessing, sensing gases and environmental parameters, and communication and data transmission. The responses of three metal oxide semiconductor sensors, BME680, ENS160, and SGP41, integrated into the sensing layer have been recorded for various volumes of selected chemicals and volatile organic compounds, including ammonia, pentane, tetrahydrofuran, butanol, phenol, xylene, benzene, ethanol, methanol, acetone, toluene, and isopropanol. For mobile applications, the sensor node was attached to a sample holder on a mobile robot (ASTI ProBOT L). In addition, the sensor nodes were positioned close to automation systems, including stationary robots. The experimental results revealed that the tested sensors have a different response to the tested volumes and can be used efficiently for hazardous gas leakage detection and monitoring.

## 1. Introduction

The great and rapid development in the field of stationary and mobile robots has enabled the provision of many attractive advantages in the work environment, such as increasing accuracy, productivity, and efficiency and reducing cost and waste of raw materials, which has made the demand for their use in factories, laboratories, and warehouses increase significantly [1,2,3]. The increasing automation of chemical and biological processes will lead to robot-based automation systems in the future that work independently and without the presence of humans. This leads to both a reduction in operational expenses and the guarantee of greater occupational safety. At the same time, there is an urgent need for monitoring systems, especially in places where hazardous materials such as radioactive materials, toxic and dangerous chemicals, and dangerous biological materials are dealt with. Recently, the leakage of ammonium nitrate in the warehouses of the port of Beirut (the capital of the State of Lebanon) led to massive destruction, which some called Beirushima [4], analogous to the nuclear explosion in the Japanese city of Hiroshima, in addition to the human losses and losses of lifeforms in the ocean (land and sea). Such horrific accidents can be avoided by using monitoring and warning systems that enable humans to assess risks, especially in places that are not in direct contact with the human factor, and where leakage risks cannot be detected by humans quickly and decisively [5].

The process of developing monitoring systems and detecting various hazardous gases and chemicals is still ongoing at a high pace, supported by the development of the production of relevant sensors [6,7]. Several efforts have been made in this direction. Schilt et al. proposed a low-cost sensor node for air quality monitoring [8]. The system consists mainly of two microcontrollers ESP8266/ESP32 (Espressif Systems, Shanghai, China) and FiPy (Pycom Ltd., Hampshire, UK) to implement parallel tasks, and several air quality sensors such as the PM3015SN particulate matter sensor (Cubic Sensor and Instrument Co., Ltd., Wuhan, China), SGP30 and SHT35 (Sensirion AG, Stafa, Switzerland) for VOCs (volatile organic compounds) and environmental parameters measurements, and OX-A431 and NO2-A43F (Alphasense, Braintree, UK) electrochemical gas sensors for O_3_ and NO_2_ measurements. Several sensor nodes have been tested with different filtering methods, and the results showed good correlation with stationary reference stations. Similar studies have been proposed in [9,10,11].

Aashiq et al. proposed an IoT-based monitoring station for air quality and environmental parameters [12]. The station is managed by two microcontrollers including the primary Arduino UNO microcontroller for data preprocessing and the secondary Node MCU Microcontroller (Espressif Systems, Shanghai, China) for data transmission to the monitoring and control station. The station has several sensors, including ZP07-MP503 (Winsen Electronics Technology, Zhengzhou, China) for VOCs concentration measurements, MQ-7 (Winsen Electronics Technology, Zhengzhou, China) for CO gas measurement, PMS 5003 (PlanTower, Nanchang, China) for 2.5 particulate matter measurements, and AHT10 (Guangzhou Aosong Electronic Co., Ltd., Guangzhou, China) and BMP280 (Bosch Sensortec, Reutlingen, Germany) for temperature, humidity, and pressure measurements. The acquired sensor data can be directly viewed using a small 0.96′ OLED display on the portable station as well, and the data will be transmitted to the ThingSpeak platform (The MathWorks, Inc., Natick, MA, USA) for data monitoring in mobile devices. Similar IoT-based approaches have been proposed in [13,14,15]. 

The widespread use of robots, unmanned vehicles, and drones has prompted many researchers to develop sensor nodes that can be integrated with this type of host and media that provide an efficient power source and advanced communication systems to transfer sensor node data to monitoring and control centers. Marques et al. proposed an air quality monitoring system using an assistive robot and the Internet of Things (IoT) technology for enhancing living environments [16]. The system used a portable sensor node consisting mainly of the MQ6 LPG gas sensor (Winsen Electronics Technology, Zhengzhou, China) and the Sun SPO wireless module (Sun Microsystems, Inc., Natick, MA, USA) attached to a TurtleBot2 mobile robot (open-source robot platform). The system can collect environmental data from several sensor nodes, which are connected to a gateway that is responsible for sending all acquired data to the robot controller (Laptop) via a USB interface port. The system can trigger an alarm and publish it directly to the social network (Facebook) using internet web services.

Chang et al. proposed a multipurpose monitoring system for industrial, medical, and military applications [17]. It is hosted by a self-balanced two-wheel mobile robot. The system has been managed by the single-board computer Raspberry Pi 3 Model B (Raspberry Pi Foundation, Cambridge, UK) with Bluetooth and wireless connectivity. The system has two sensors for chemical vapors and gases detection, which are the MEMS-based Grove-MICS6814 (SGX Sensortech Limited, Neuchatel, Switzerland) and the MOX-based MQ-2 (Winsen Electronics Technology, Zhengzhou, China), which allow the system to detect the leakage of several gases such as NO_2_, C_3_H_8_, C_4_H_10_, CH_4_, NH_3_, CO, C_2_H_5_OH, Smook, and LPG. All the sensors’ data can be transferred to the monitoring station via an IoT cloud. Similar robot-based studies have been proposed in [18,19,20,21,22,23]. 

Park et al. proposed a drone-based gas detection system for remote monitoring [24]. The gas sensing element is fabricated from a graphene chemiresistor array. The system used the commercial IoT-based Heltec WiFi LoRa 32 (Heltec Automation, Chengdu, China) development board for gas sensor data processing and transitions. The system used the BME280 (Bosch Sensortec, Reutlingen, Germany) environmental sensor for air pressure, temperature, and humidity data acquisition. The system has been successfully tested for ethanol vapor, humidity, and temperature, also the measured data can be monitored within a 1 km distance. Similar approaches have been proposed in [25]. 

In this paper, the development and testing of a portable sensor node to detect and monitor the leakage of hazardous chemical gases and vapors during robotics-based operations is presented. The sensor node works with the Internet of Things technology to transfer measured data to database servers and live monitoring and control stations. This node is characterized by unique characteristics, such as the great flexibility in changing the sensing layer according to the type of gas to be monitored. It also allows direct live monitoring by displaying the measured data on a digital screen covering the top surface of the node. The node has been tested with stationary and mobile robots in different scenarios. The first scenario was performed in a specific chemical preparation hood at two distances between the chemicals and the sensor node, which is used with stationary robots. In the second scenario, the node attached to the ASTI ProBOT L mobile robot (ASTI Mobile Robotics, Burgos, Spain) which is programmed for chemical and biological sample transportation in the laboratories of the Center for Live Science Automation (Celisca), University of Rostock, Germany. The sensor node was tested with several selected chemicals and volatile organic compounds (VOCs) to evaluate the responses of three different gas sensors to different volumes of the tested substances. In comparison to previous studies in the literature, the proposed node has the following advantages: The multi-sensor sensing layer provides the ability to detect leakages of a wide range of chemicals, as confirmed by the tests.The possibility of displaying the recorded data directly on the node itself using the built-in display screen, in addition to the possibility of viewing the data remotely through the monitoring program in the control stations through the cloud.The presence of a Bluetooth module in the communication layer provides the possibility of using the node to track the location of a chemical leakage using Beacon Bluetooth technology.

The paper is divided as follows: The second section includes an introduction to the materials and parts used in the sensor node. In the third section, the mechanism of tests, the conditions used, and the results achieved for each test are reviewed. In the fourth section, the results and their implications are discussed. Finally, in the fifth section, the conclusions and the possible future research are presented. 

## 2. Materials and Methods

The sensor node consists mainly of four layers combined with a rechargeable battery. The sensor node components are covered with a colorless acrylic material to protect them from direct exposure to any factors that affect their performance, such as liquids or chemicals falling directly on sensors and other electronic elements, especially when used with mobile robots to transport various types of samples in automated laboratories. Figure 1 explains the internal structure and the main layers of the developed sensor node.

The first layer is located at the bottom of the sensor node and is called the power management layer. This layer regulates the electrical voltage for the sensor node, which can be of different voltages depending on the nature of the host. The sensor node can be fed with a continuous voltage of 24 volts, especially if it is used in fixed places or with mobile and stationary robots that operate with electrical voltages approaching this level. It can also be fed with a low voltage of 3.7 volts from rechargeable lithium-ion batteries built into the sensor node if used in places where no external electrical power is available. The use of this layer can be omitted if there is an input to the Type-C connection interface from any external host that supplies power to the sensor node and charges the node battery. 

The second layer that sits above the first layer is the central processing layer. This layer contains the microcontroller responsible for managing the work of the sensor node. This microcontroller has been programmed to carry out the initial processing of the data of all sensors in the sensor node, which are connected to it through different types of data buses such as UART, SPI, I2C, and USB Type C interfaces. It also organizes the transmission of information from the sensor node to the monitoring and control station through the communication layer that sends the specific information required by the users through the cloud. The NXP MKL27Z256LH4 microcontroller (NXP Semiconductors, Eindhoven, the Netherlands) was selected to be the brain of the Celisca sensor node. It is a 32-bit ARM-based controller with a processing clock of 32 MHz. The processing layer contains the MCP79411 (Microchip Technology Inc., Chandler, AZ, USA) real-time clock. It is mainly used to generate timestamps to document the direct readout of the sensing layer data before it is sent from the sensor node.

The sensing layer is located on top of the processing layer. This layer consists of several sensors, including sensors for gases and chemical vapors as well as sensors for environmental factors such as pressure, temperature, and humidity. The used sensors are BME680 (IAQ, TVOC, CO_2_, Temperature, Humidity, Pressure), SGP41 (VOC-Index, NOx-Index), ENS160 (IAQ, TVOC, CO_2_), PGS1004 (H_2_), and the pressure sensor MS5803-05BA (TE connectivity, Schaffhausen, Switzerland). All sensors in the sensing layer send the measured data to the processing layer via the available I2C, SPI, and UART communication buses based on the sensor communication protocol. Table 1 provides details of the used sensors and their general specifications. 

The communication layer is located on top of the sensing layer. There are two main functions for this layer. The first is the process of transferring data from the sensor node to the cloud using a Wi-Fi communication unit to deliver data to monitoring and control stations using the Wi-Fi module ESP-WROOM-02D (Espressif Systems, Shanghai, China). The second is the process of determining the location of the sensor node using the ACN52840 (Aconno, Düsseldorf, Germany) Bluetooth module, which transmits the location to a network of beacon units installed on the walls of the building. 

In addition to sending data to monitoring and control stations, the sensor node data can also be viewed directly on a small 480 × 280 pixels Pico-ePaper-3.7 inch (9.4 cm) black and white screen (Waveshare Electronics Co. Ltd., Shenzhen, China), which is placed on the top of the sensor node (see Figure 2). It is an ultra-low power consumption screen that can keep showing the last content on it as pixels for a long time even when power is shut. The E-paper screen communicates with the sensor node via the SPI interface and is powered by 3.3 volts from the sensor node power management layer. It has an acceptable full refresh time of ≈3 s for updating viewed data. Figure 2 shows the tested Celisca sensor node.

The sensor node layers can be controlled directly via USB Type-C interface or IoT cloud using a special graphical user interface (GUI). The GUI can be run from any computer or tablet equipped with the MS Windows operating system, and it can be run from several computers at the same time. The GUI allows the user to choose between sensor nodes to display data from different places using the unique serial number assigned to each sensor node. By the same mechanism, the user can display data previously stored in the database. Furthermore, it allows the user to select any specific sensor or parameter from the chosen sensor, change the measurement sample rate, sensor calibration parameters, and store the sensor data. Figure 3 shows the used GUI of the sensor node’s data collection software. 

The ASTI ProBOT L mobile robot was used as the host of the developed sensor node through the second testing phase. This robot’s dimensions (L × W × H) are 734 × 630 × 1515 mm, and it moves in bidirectional mode. The robot is designed to transport medium loads (up to 50 kg) with speeds ranging from 0.1 to 1.2 m/s. It provides reliable and high-quality solutions for automating transportation. The robot works with the latest SLAM advanced navigation technologies for mobile robots, which help in flexible and dynamic planning and secure collaboration with operators in the work environment [31]. The external sensor housing is designed in the standard SBS format and fits onto the sample tray with nine sample positions (see Figure 4). 

## 3. Experimental Tests and Results

The sensor node tests were conducted in a laboratory containing an air ventilation system programmed to maintain a stable temperature of around 22.0 ± 0.5 °C and a relative humidity of around 50.0 ± 2.0%. The following section explains the tests and results for mobile robot and stationary robot applications.

### 3.1. Tests and Results for Mobile Robots

The main purpose of these tests was to record and evaluate the response of the sensor node to the leakage of hazardous chemical substances and gases while working in the open laboratory environment, especially when transporting chemicals by mobile robots or other automated transport operations. Thus, the sensor node was tested with five selected widely used VOCs substances (ethanol, methanol, acetone, toluene, and isopropanol). Each substance was tested five times with different testing sample volumes (50, 100, 300, 500, and 1000 µL). The solvents were injected into a 15 cm diameter Petri dish located approximately 75 cm aside from the mobile robot path. The robot started from an initial point, passed through a narrow corridor, and then returned to the starting point. The sample was injected into the Petri dish when the robot started the movement. The test results are recorded directly using a laptop via a USB Type-C cable, and at the same time, the sensor node sends the recorded data wirelessly to a central monitoring and control center via the cloud. Figure 5 shows the test configuration in the laboratory with the mobile robot path.

The testing path has been selected in one of the automated laboratories where real-time automated chemical sample transportation tasks are carried out frequently. This narrow path is typical for biological, chemical, or analytical laboratories. These corridors are used by the staff and the mobile robots for sample transportation. Figure 6,Figure 7,Figure 8,Figure 9,Figure 10 explain the responses of the sensors BME680, SGP41, and ENS160 for the materials injected into the Petri dish and in the chosen volumes.

The process of detecting leakages of gases and chemical vapors in a relatively large space, such as a laboratory or warehouse, is crucial in quickly avoiding potential risks resulting from the leaks. In the previous testing, four parameters were chosen and measured from the three gas sensors in the sensing layer to detect the leakage of gases and chemical vapors, which are as follows: IAQ-index and TVOC from BME680, and VOC-index from SGP41, and TVOC from ENS160. The results shown in Figure 6,Figure 7,Figure 8,Figure 9,Figure 10 revealed a good, logical, and consistent response for the chosen test volumes, and the results achieved can be summarized as follows:BME680: Tests indicate that the sensor was capable of effectively detecting ethanol leakages with a volume of ≥300 µL. In contrast, the sensor’s response was weak for all test samples for acetone and toluene. The sensor’s response was also moderate for isopropanol; effective responses were reached for samples of ≥100 µL. The sensor’s response was also acceptable for methanol samples, as it was able to clearly detect leakage for samples of ≥300 µL.SGP41: Tests showed that the sensor was capable of effectively detecting ethanol leakages with a volume of ≥100 µL. The sensor’s response was good for acetone samples of ≥300 µL as well as for all tested isopropanol volumes (effective response for samples of ≥50 µL). Furthermore, the sensor showed an excellent response (the best compared to the rest of the tested materials) to all tested sizes of methanol, as it was able to clearly detect leakage for samples of ≥50 µL. In addition, sensor responses were also good for toluene samples with a volume of ≥100 µL.ENS160: The sensor showed good response to the tested sample volume of ≥300 µL of ethanol. In contrast, the sensor’s response was weak for all test samples for acetone, isopropanol, and toluene. The sensor’s response was also weak for methanol samples, as it was able to clearly detect leakage for samples of ≥1000 µL.

The evaluation of the test results shows that the BME680 sensor was unable to detect any sample of acetone or toluene. Furthermore, the ENS160 sensor revealed the weakest response, and it failed to detect all tested samples except ethanol and methanol. The SGP41 sensor shows the best performance and can detect all the tested substances. 

The ability of a single sensor to detect different materials depends on the type of sensitizing material in the internal sensor structure, which is difficult to modify in commercial sensors. Therefore, several sensors were used together in the sensing layer to avoid the failure of one of the sensors and to ensure an effective increase in the spectrum of chemical substances that can be detected.

### 3.2. Tests and Results for Stationary Robots Applications

The second phase of testing for stationary robot application was performed in a laboratory equipped with a Secuflow fume hood (Waldner Holding GmbH & Co. KG, Wangen im Allgäu, Germany), which has a special ventilation system to avoid the spreading of gases and chemical vapors in the vicinity. The ventilation system is obligatorily turned off during the test period (an average of 10–15 min) to obtain a natural response to the gas diffusion without affecting the ventilation system. It is then turned on to empty the hood before subsequent testing is carried out. A setup was chosen for experiments that simulate the use of sensor nodes with mobile robots, where two heights of 40 cm and 100 cm between the node and the leakage sources were adopted based on previous tests [3,7], which revealed an inverse proportion relation between the distance and the sensor responses. Thus, the easy detection of leakage from the level of the laboratory floor and from the level of stations carrying chemicals to be transported is achieved. The distance between the sensor node and the test model is controlled by installing the sensor node on a metal stand that can be adjusted in height. Seven different chemicals were chosen to evaluate the performance of the sensor node, including ammonia, pentane, tetrahydrofuran, butanol, phenol, xylene, and benzene. These solvents are typical compounds used in various laboratories and industry, and they have not been tested previously. In addition, these components are also included in many household products for cleaning, sterilization, etc. The concentration of all chemicals was 100% except ammonia which was 25%. Five volumes (50, 100, 300, 500, and 1000 µL) are tested for the selected chemicals. Phenol was tested with only three volumes (50, 100, 300 µL) due to the long recovery time required for all sensors with this substance. The samples are placed in a custom glass Petri dish using a special Eppendorf micropipette (Eppendorf, Hamburg, Germany). Each sample was tested twice with the two selected distances. Figure 11 shows the experimental setup.

Before each test, it is considered that the baseline for the indoor air quality (IAQ) in the hood is within acceptable limits that do not exceed 100 degrees (0 ≤ IAQ ≤ 500). If this limit is exceeded, the ventilation system in the hood is operated to obtain acceptable air quality for conducting tests. During the test, the response of the three gases and chemical vapor sensors (BME680, SGP41, and ENS160) have been recorded for each substance and all concentrations from the predetermined distances (40 cm and 100 cm). As each sensor can measure several parameters, the results in this paper concentrate on one parameter from each sensor, and the parameters are as follows: the total volatile organic compounds (TVOCs) concentration measured in part per million for BME680 and ENS160 sensors and the volatile organic compound index (VOC-index) for the SGP41 sensors. 

The test results were captured directly from the sensor node and stored on a laptop via a Type-C data cable using the GUI, which provides the ability to store the measured data as a comma-separated values file (csv) that can easily converted to an Excel sheet. The data sent from the sensor node is recorded for effective periods that start from the normal readings of the sensors within the limits of the baseline, then the test material is added inside the hood directly under the sensor node, then the response of the sensors is awaited to reach the maximum peak (peak value), and also sensors are awaited to return to the baseline. In some tests, the sensors showed long recovery time (>30 min), to complete the test and return to the baseline, so it is often sufficient to record the response for 20 min. Figure 12,Figure 13,Figure 14,Figure 15,Figure 16,Figure 17,Figure 18 show the response of BME680, SGP41, and ENS160 sensors for the tested chemicals from the two 40 cm and 100 cm distances. 

The data presented in Figure 12,Figure 13,Figure 14,Figure 15,Figure 16,Figure 17,Figure 18 for several selected chemical substances provide a clear picture of the extent to which the response of the sensors differs for each substance in terms of the possibility of detecting the substance or not, as well as the strength of the response. The different responses to different materials make it necessary to use a larger number of sensors in the sensor node when the number of materials that can leak into the area to be monitored increases. The test results in the previous figures can be clarified as follows: **Ammonia**: The tests indicate that the BME680 sensor can detect volumes of ≥50 µL, with a maximum response (saturation = 1000 ppm) for a 40 cm distance. The response is less when increasing the distance to 100 cm, where only volumes of ≥500 µL can be detected. The ENS160 sensor can detect volumes of ≥100 µL, with a maximum response of ≈9.5 ppm for a 40 cm distance, and the response is lower when increasing the distance to 100 cm, where only volumes of ≥500 µL can be detected. The SGP41 sensor detects volumes of ≥500 µL, with a maximum response of ≈200 VOC-index for a 40 cm distance, and only volumes of ≥1000 µL are detected for the test distance of 100 cm.**Benzene:** The tests prove that the BME680 sensor has a weak response and can detect only volumes of ≥500 µL, with a maximum response of ≈600 ppb at a 40 cm distance. Increasing the distance to 100 cm resulted in a lower response; thus, no volumes could be detected. The ENS160 sensor was unable to detect any volumes for both distances. The SGP41 sensor shows an excellent response for both distances. It can detect volumes of ≥50 µL at a 40 cm distance, with a good response for volumes of ≥100µL. For a distance of 100 cm, measurement volumes of ≥100µL can be detected.**Butanol:** The tests strongly suggest that the BME680 sensor can detect volumes of ≥100 µL, with a maximum response of ≈2.5 ppm for 1 mL volume at a 40 cm distance. The response reduces when increasing the distance to 100 cm, where it can detect only volumes of ≥500 µL. The ENS160 sensor cannot detect any volumes for both distances. The SGP41 sensor shows a good response for volumes of ≥100 µL at a 40 cm distance, and a response for volumes of ≥300 µL for the test distance of 100 cm.**Pentane:** The tests show that the BME680 sensor can only detect volumes of ≥300 µL for a 40 cm distance. For a distance of 100 cm, only volumes of ≥1000 µL can be detected. The ENS160 sensor cannot detect any volumes for both distances. The SGP41 sensor can detect volumes of ≥100 for a 40 cm distance, and it detects only volumes of ≥300 µL for the test distance of 100 cm.**Phenol:** The tests indicate that the BME680 sensor can detect all tested volumes of ≥50 µL for a 40 cm distance. The response is less when increasing the distance to 100 cm, where it can detect only volumes of ≥100 µL. The ENS160 sensor cannot detect any volumes for both distances. The SGP41 sensor shows good responses and can detect all the tested volumes of ≥50 µL for both distances. The sensor recovery time for this compound is very long (around 30 min), so only three volumes (50 µL, 100 µL, 300 µL) were tested for phenol.**Tetrahydrofuran:** The tests indicate that all sensors can detect all the tested volumes ≥50 µL for both distances.**Xylene:** The tests indicate that the BME680 sensor can detect volumes of ≥300 µL for a 40 cm distance, and volumes of ≥500 µL for a 100 cm distance. The ENS160 sensor can detect volumes of ≥500 µL for a 40 cm distance and cannot detect any volumes for a 100 cm distance. The SGP41 sensor can detect all the tested volumes of ≥50 µL for a 40 cm distance, and it detects only volumes of ≥100 µL for the test distance of 100 cm.

The results of the sensor node tests are summarized in Table 2, where the recorded response represents the TVOC in ppm for both the BME680 and ENS160 sensors and the VOC index (0–500) for the SGP41 sensor. 

The step response time T90 and recovery time were calculated for both the tested sensors. The step response time is defined as the time it takes for the sensor to reach 90 percent of its maximum reading, while the recovery time can be defined as the time it takes for the sensor to return from the highest value recorded to the previous reading level before the start of the test (baseline reading level). Ammonia and tetrahydrofuran at a concentration of 1000 mL were chosen to calculate the step response and recovery times. These materials were chosen due to the presence of comprehensive data to calculate the recovery time within the time used for the tests (15–20 min), as some materials take longer than that. Figure 19a,b explain the T90 step response time and the recovery time for the tested sensors with ammonia and tetrahydrofuran, respectively.

The results in Figure 19 show that the SGP41 sensor has the shortest T90 step response time (111 s), while the ENS160 sensor has the longest T90 time (356 s). Moreover, the BME680 sensor has the shortest recovery time (425 s), while the SGP41 sensor has the longest recovery time (1388 s). Calculating the recovery time is critical for monitoring and warning applications, since it determines the inactive period during which the sensor cannot make useful measurements. The recorded response and recovery times confirm the need to have more than one sensor in the sensor nodes to provide sufficient flexibility to re-detect any new leakages that may be of another substance, with a short time interval. In all cases, the recorded response and recovery times are considered practical for the operation of the sensor node, as the leakage warning will remain continuous until all sensors return to the baseline reading level. In previous tests, the sensor node was tested with only one type of material in each test, but not for its response to a mixed leak of several gases or chemicals. The tested sensors, BME680, SGP41, and ENS160, use metal oxide technology (MOX) where the resistance of the sensing elements changes based on the targeted gas concentration. 

The results achieved in this study are different from those of previous studies present in the literature as a wide range of chemicals and volatile materials were tested under different test conditions suitable for the automated work environment using fixed and mobile robots. In each test, the node was tested for several volumes of sample from the smallest of 50 µL and increased to 100 µL, 300 µL and 500 µL until the largest testing volume of 1000 µL for all tested material for both stationary and mobile robots. This revealed the minimum leakage volume of the tested samples that can be detected by every single sensor used in the node. 

## 4. Conclusions and Future Research

The problem of leakage of chemical materials and gases requires early detection and rapid response to avoid any possible damage to lives and property. In this paper, the components and design of a portable sensor node designed for integration with stationary and mobile robots used in chemical and biological material transport operations are reviewed. The performance of the sensor node was tested and evaluated by testing its ability to detect leakage of several types of chemicals, considering conditions that simulate the use of the sensor node in fixed places and when attached to mobile robots in terms of the distance between the sensor node and the source of the leak. The focus was on the response of three commercial sensors for gases and chemical vapors integrated into the sensor node and comparing their response to selected materials. Experiments have proven that the response of the sensors to the tested samples varies in terms of whether they can detect a leak and the strength of the response signal. Therefore, using more than one sensor provides the sensor node greater flexibility in confirming the leak detection of target materials to avoid any errors when one of the sensors is not responding to any potential leak. The results also indicated that the performance of the BME680 and SGP41 sensors is better than that of the ENS160 sensor for most of the tested materials. Therefore, in future research, it is possible to replace this sensor or add other sensors to the sensing layer in the sensor node to expand the spectrum of materials that can be detected for leakage, according to the commonly used materials in the target laboratory/location. 

## Figures and Tables

**Figure 1 sensors-24-01295-f001:**
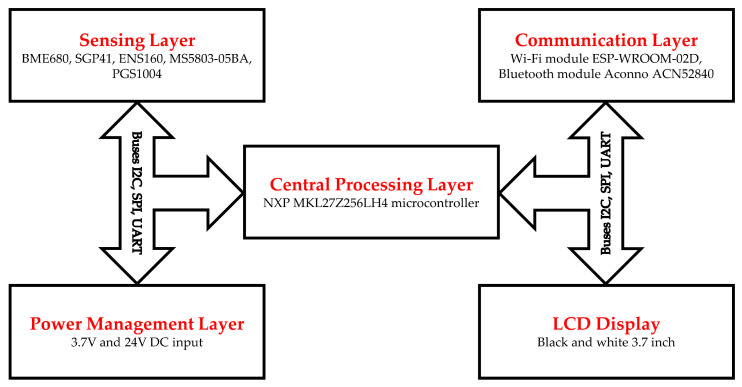
Internal structure of the sensor node with main layers.

**Figure 2 sensors-24-01295-f002:**
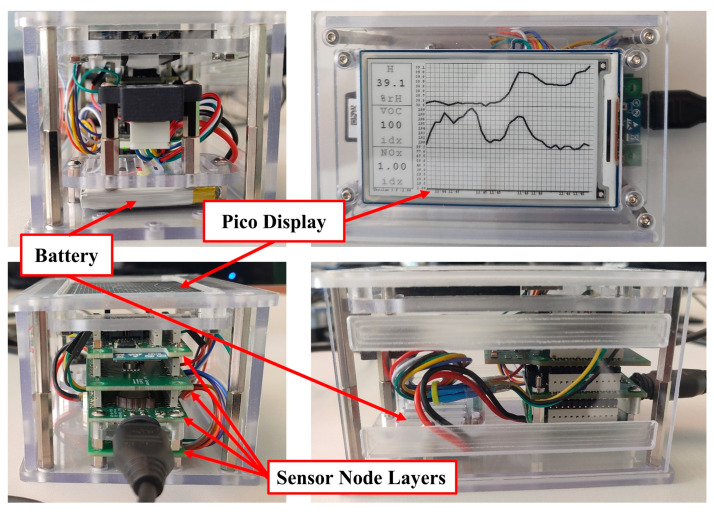
Celisca sensor node.

**Figure 3 sensors-24-01295-f003:**
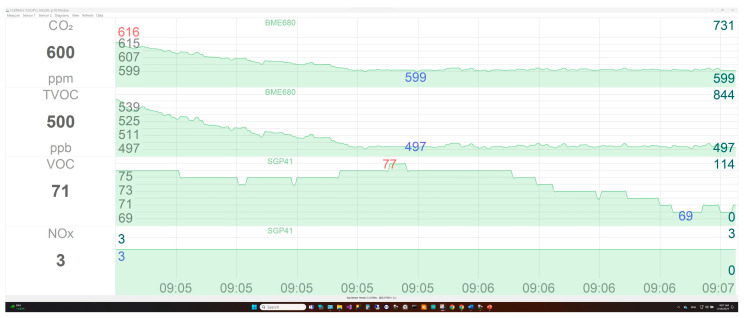
Graphical user interface of the sensor node software.

**Figure 4 sensors-24-01295-f004:**
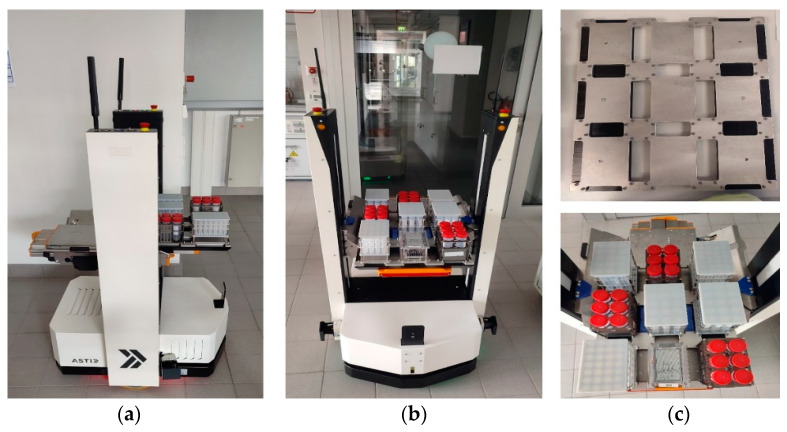
The ASTI ProBOT robot: (**a**) side view, (**b**) front view, and (**c**) robot sample tray.

**Figure 5 sensors-24-01295-f005:**
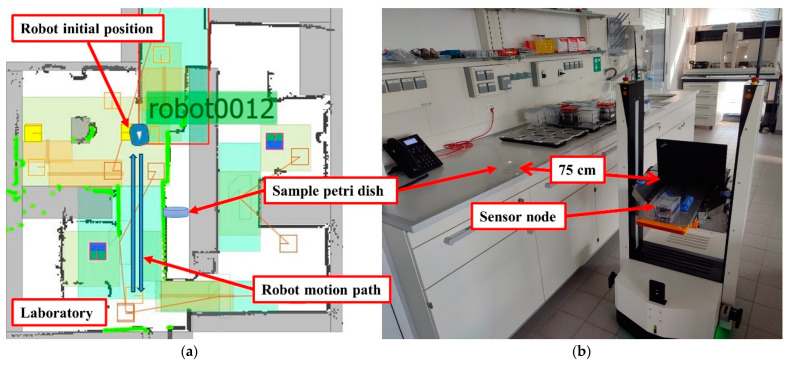
The testing environments: (**a**) screen shoot of robot path planning software and (**b**) real-time test path.

**Figure 6 sensors-24-01295-f006:**
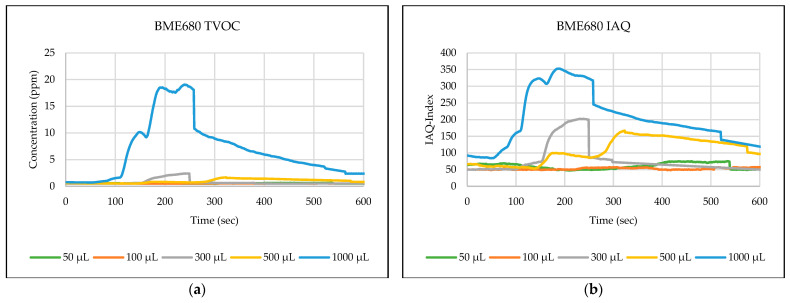
Sensors’ test results for ethanol: (**a**) BME680 TVOC, (**b**) BME680 IAQ, (**c**) SGP41 VOC, and (**d**) ENS160 TVOC.

**Figure 7 sensors-24-01295-f007:**
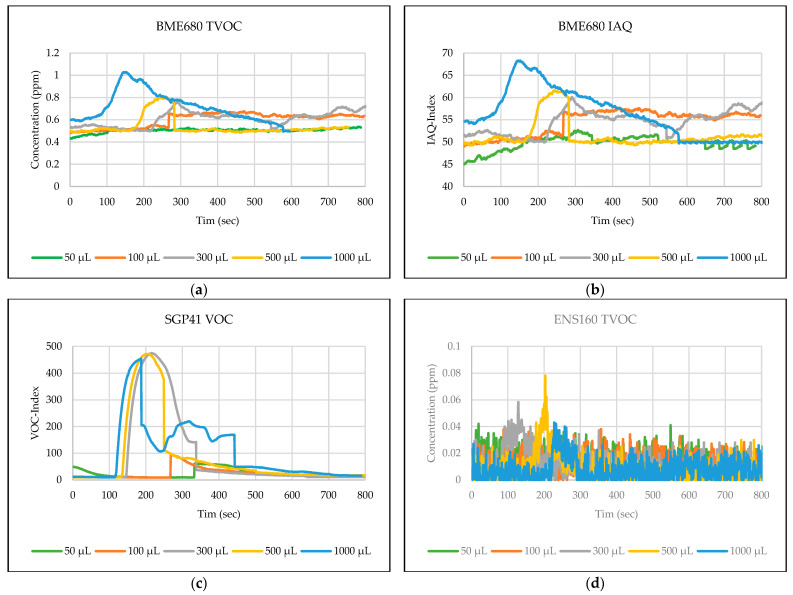
Sensors’ test results for acetone: (**a**) BME680 TVOC, (**b**) BME680 IAQ, (**c**) SGP41 VOC, and (**d**) ENS160 TVOC.

**Figure 8 sensors-24-01295-f008:**
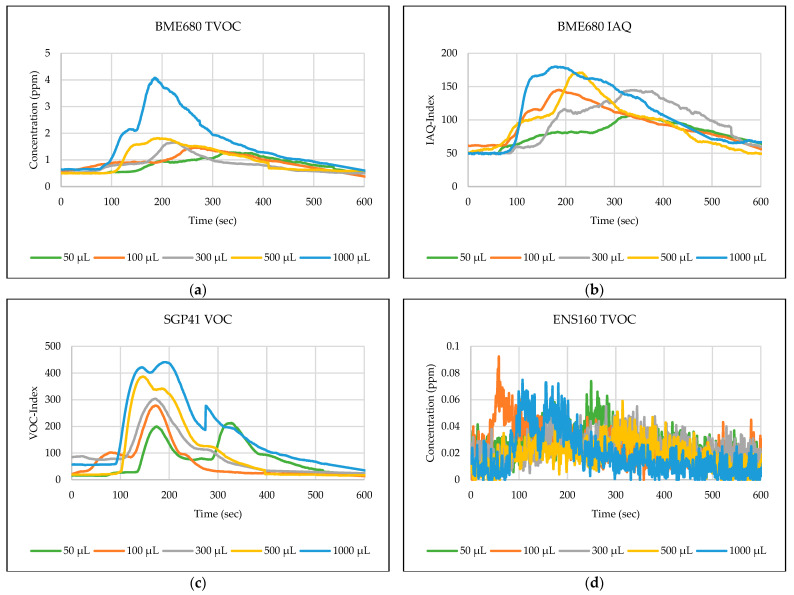
Sensors’ test results for isopropanol: (**a**) BME680 TVOC, (**b**) BME680 IAQ, (**c**) SGP41 VOC, and (**d**) ENS160 TVOC.

**Figure 9 sensors-24-01295-f009:**
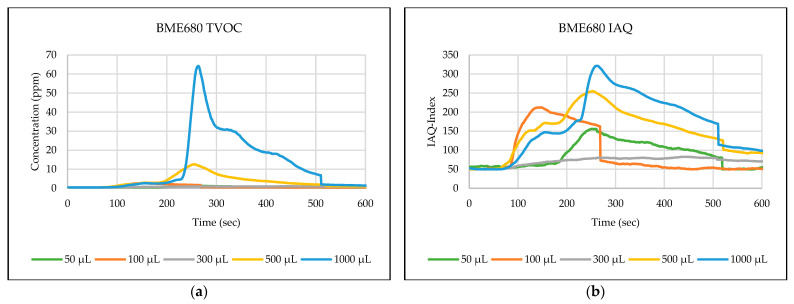
Sensors’ test results for methanol: (**a**) BME680 TVOC, (**b**) BME680 IAQ, (**c**) SGP41 VOC, and (**d**) ENS160 TVOC.

**Figure 10 sensors-24-01295-f010:**
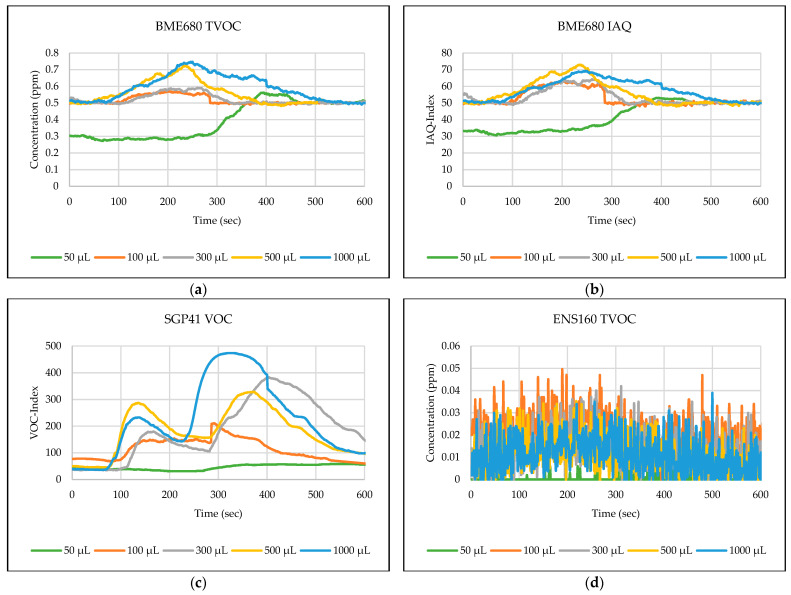
Sensors’ test results for toluene: (**a**) BME680 TVOC, (**b**) BME680 IAQ, (**c**) SGP41 VOC, and (**d**) ENS160 TVOC.

**Figure 11 sensors-24-01295-f011:**
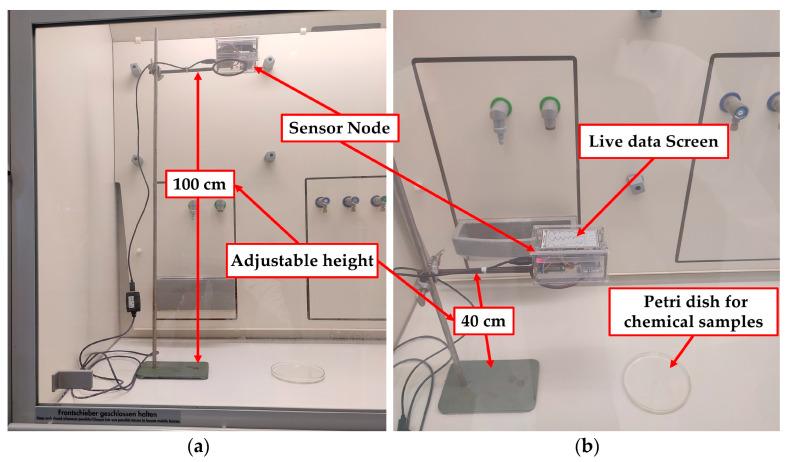
Experimental setup for chemical testing: (**a**) 100 cm and (**b**) 40 cm.

**Figure 12 sensors-24-01295-f012:**
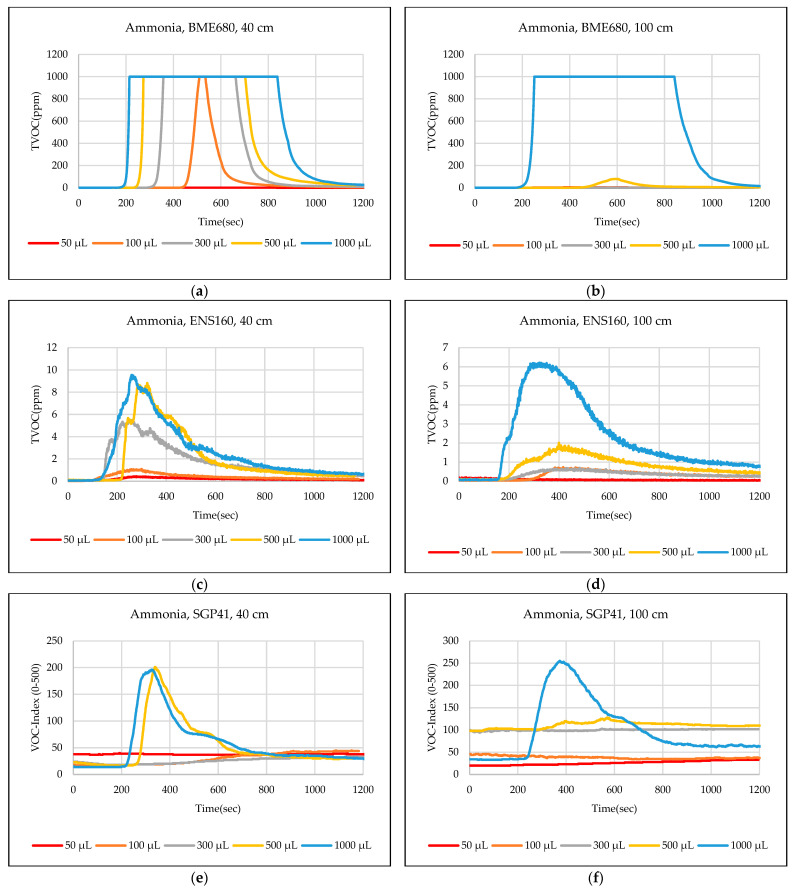
Sensors’ test results for ammonia: (**a**) BME680, 40 cm; (**b**) BME680, 100 cm; (**c**) ENS160, 40 cm; (**d**) ENS160, 100 cm; (**e**) SGP41, 40 cm; and (**f**) SGP41, 100 cm.

**Figure 13 sensors-24-01295-f013:**
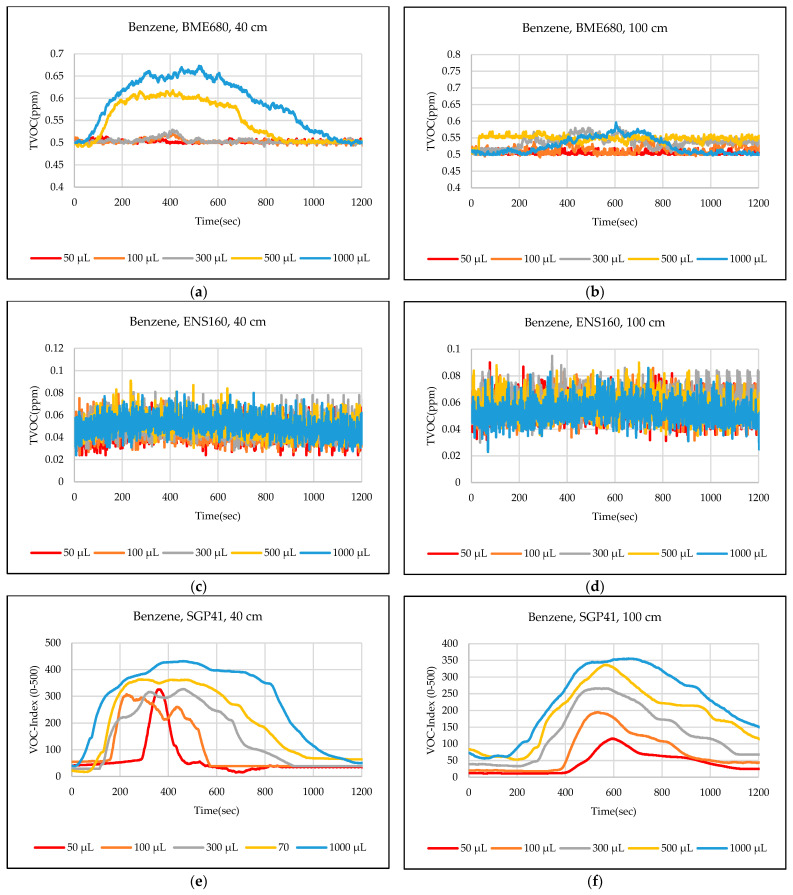
Sensors’ test results for benzene: (**a**) BME680, 40 cm; (**b**) BME680, 100 cm; (**c**) ENS160, 40 cm; (**d**) ENS160, 100 cm; (**e**) SGP41, 40 cm; and (**f**) SGP41, 100 cm.

**Figure 14 sensors-24-01295-f014:**
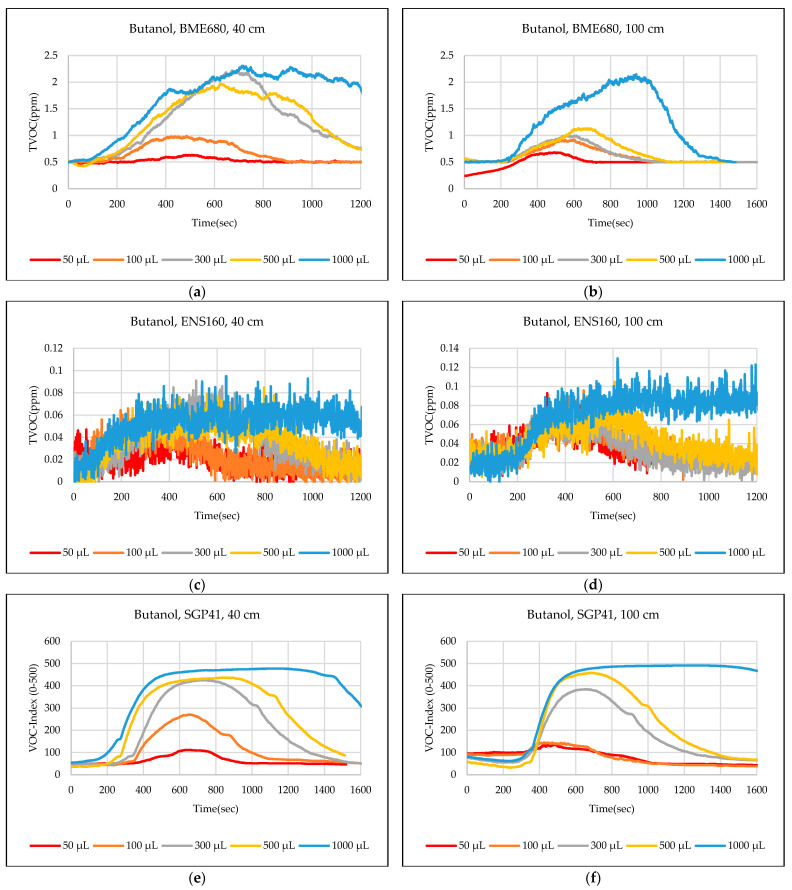
Sensors’ test results for butanol: (**a**) BME680, 40 cm; (**b**) BME680, 100 cm; (**c**) ENS160, 40 cm; (**d**) ENS160, 100 cm; (**e**) SGP41, 40 cm; and (**f**) SGP41, 100 cm.

**Figure 15 sensors-24-01295-f015:**
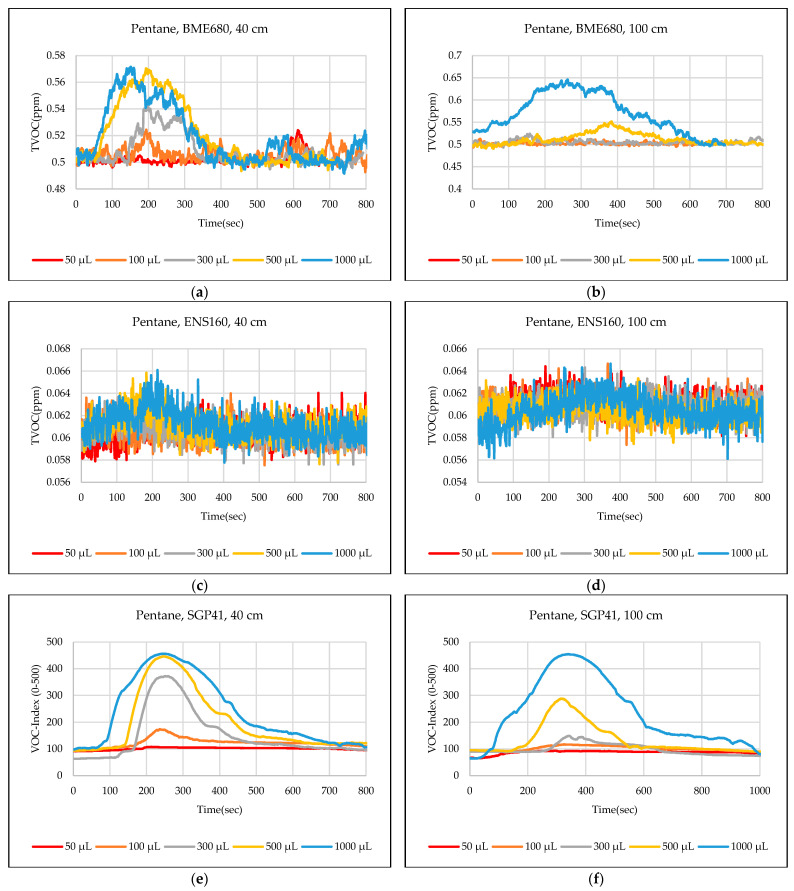
Sensors’ test results for pentane: (**a**) BME680, 40 cm; (**b**) BME680, 100 cm; (**c**) ENS160, 40 cm; (**d**) ENS160, 100 cm; (**e**) SGP41, 40 cm; and (**f**) SGP41, 100 cm.

**Figure 16 sensors-24-01295-f016:**
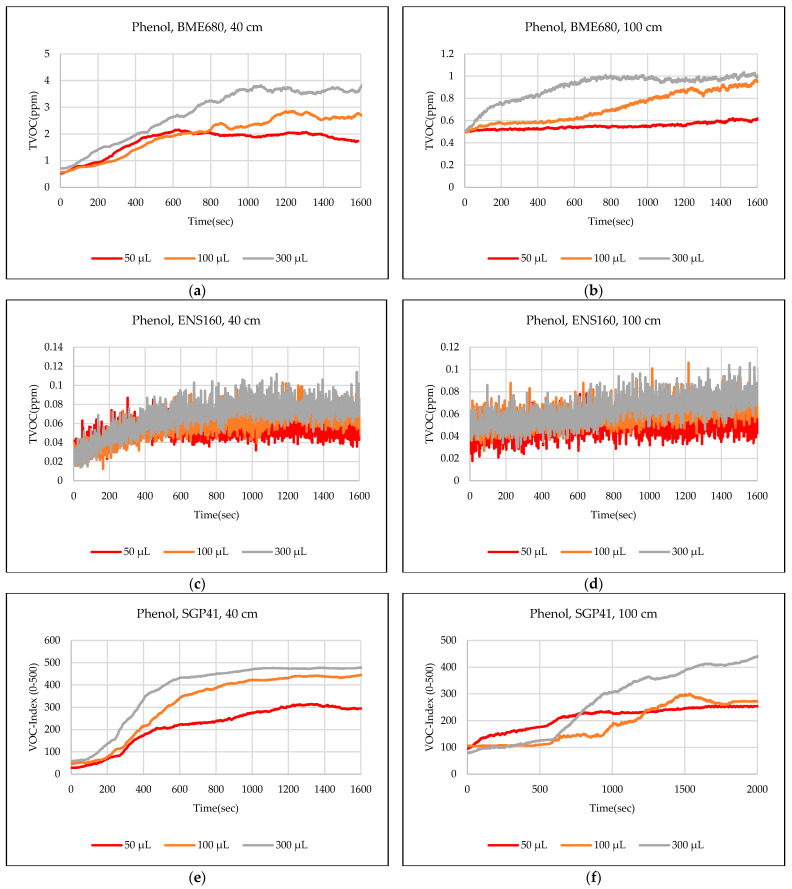
Sensors’ test results for phenol: (**a**) BME680, 40 cm; (**b**) BME680, 100 cm; (**c**) ENS160, 40 cm; (**d**) ENS160, 100 cm; (**e**) SGP41, 40 cm; and (**f**) SGP41, 100 cm.

**Figure 17 sensors-24-01295-f017:**
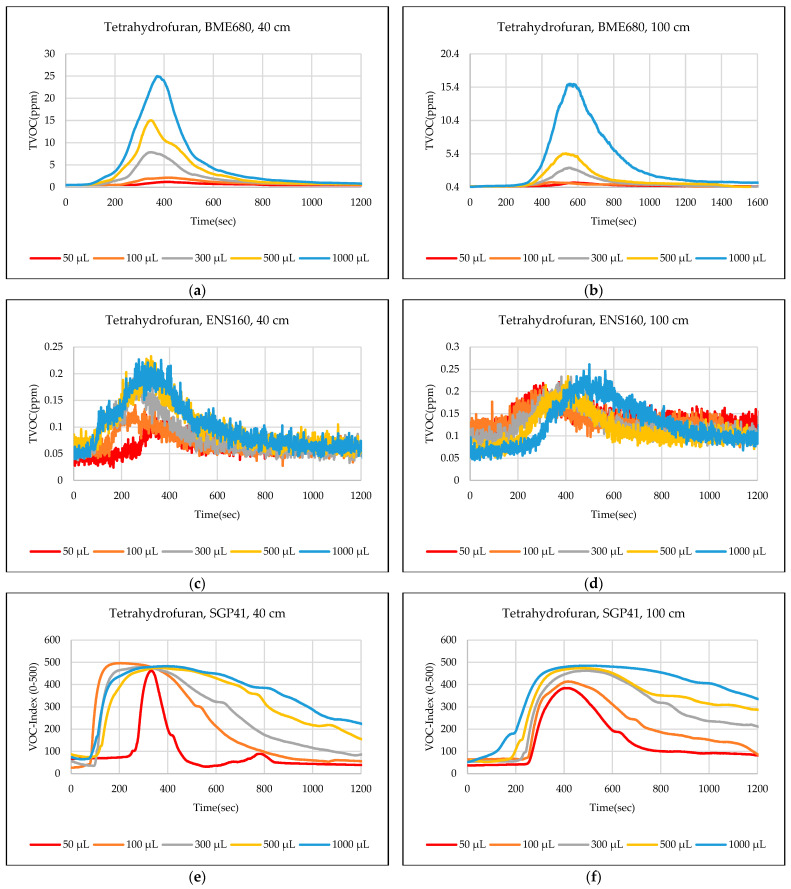
Sensors’ test results for tetrahydrofuran: (**a**) BME680, 40 cm; (**b**) BME680, 100 cm; (**c**) ENS160, 40 cm; (**d**) ENS160, 100 cm; (**e**) SGP41, 40 cm; and (**f**) SGP41, 100 cm.

**Figure 18 sensors-24-01295-f018:**
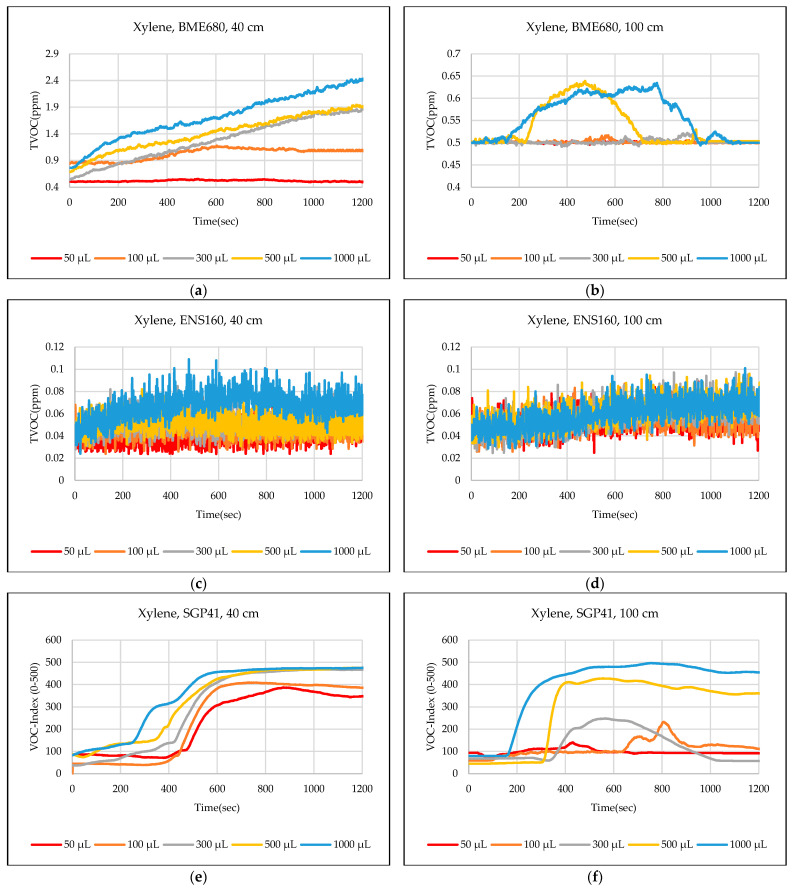
Sensors’ test results for xylene: (**a**) BME680, 40 cm; (**b**) BME680, 100 cm; (**c**) ENS160, 40 cm; (**d**) ENS160, 100 cm; (**e**) SGP41, 40 cm; and (**f**) SGP41, 100 cm.

**Figure 19 sensors-24-01295-f019:**
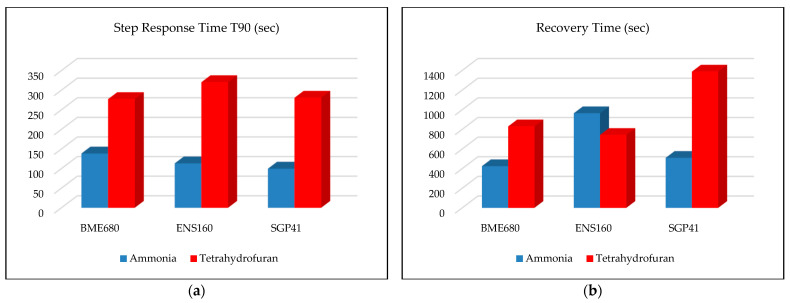
(**a**) Step response time T90 and (**b**) recovery time.

**Table 1 sensors-24-01295-t001:** The sensing layer sensors.

Sensor	BME680 [26]	SGP41 [27]	ENS160 [28]	PGS1004 [29]	MS5803-05BA [30]
Supply (V)	1.7–3.6	1.7–3.6	1.8–3.6	5	1.8–3.6
Parameters	IAQ, TVOC, CO_2_, temperature, relative humidity, pressure	IAQ-Index, NOx-Index	IAQ, TVOC	Hydrogen in air	Atmospheric pressure, temperature
Buses	I^2^C and SPI	I^2^C	I^2^C and SPI	I^2^C	I^2^C and SPI
Size (mm^3^)	3.0 × 3.0 × 0.93	2.44 × 2.44 × 0.9	3.0 × 3.0 × 0.9	23 × 20 × 10	6.4 × 6.2 × 2.88
Response time	1 s	1 s	1 s	1.4 s	1 s
Structure	MEMS	MOX	MOX	MEMS	MEMS
Manufacturer	Bosch Sensortec GmbH	Sensirion AG	Sciosense B.V.	Posifa Technologies	TE connectivity

**Table 2 sensors-24-01295-t002:** Sensor node tests results inside the hood for 40 cm and 100 cm distances.

Substance	BME680 40 cm	BME680 100 cm	ENS160 40 cm	ENS160 100 cm	SGP41 40 cm	SGP41 100 cm
Ammonia	≥50 μL	≥500 μL	≥100 μL	≥500 μL	≥500 μL	≥1000 μL
Benzene	≥500 μL	—	—	—	≥50 μL	≥100 μL
Butanol	≥100 μL	≥500 μL	—	—	≥100 μL	≥300 μL
Pentane	≥300 μL	≥1000 μL	—	—	≥50 μL	≥50 μL
Phenol	≥50 μL	≥100 μL	—	—	≥50 μL	≥50 μL
Tetrahydrofuran	≥50 μL	≥50 μL	≥50 μL	≥50 μL	≥50 μL	≥50 μL
Xylene	≥300 μL	≥500 μL	≥500 μL	—	≥50 μL	≥100 μL

## Data Availability

The data presented in this study are available on request from the corresponding author.

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
