# Peer review of "Ambient Monitoring Portable Sensor Node for Robot-Based Applications"

_sensors, 2024, doi:10.3390/s24041295_

Round 1

Reviewer 1 Report

Comments and Suggestions for Authors

In this paper, the author designs and constructs an integrated portable sensor for fixed and mobile robots in material transportation operations. Simulate two states, fixed and mobile, for experimental detection to verify the feasibility of this study. Although this study has some innovation and feasibility, there are still areas for improvement in this article.

1. The introduction section reviews the research progress related to this article in terms of monitoring and detecting chemical substances and integrating host and media design. Finally, the differences between this study and previous research should be pointed out, and the innovation points of this article should be emphasized.

2. Relevant reasons should be provided for the selection and concentration settings of seven chemical substances, namely ammonia, pentane, tetrahydrofuran, butanol, phenol, xylene, and benzene, for the application testing of fixed robots.

3. It is recommended to compare the experimental testing and results with the relevant research results at the end, highlighting the good performance of this study.

4. The title of the test result image of the sensor should be removed from the "Index" to maintain consistency among all image titles.

5. It is recommended to appropriately delete the introduction section.

Author Response

Dear reviewer, We appreciate your devoted time in reviewing our paper and your valuable comments which enabled improvements in the current version of the manuscript. The authors have carefully considered all comments and tried our best efforts to address every one of them. A point-to-point response is provided below.: 1. The introduction section reviews the research progress related to this article in terms of monitoring and detecting chemical substances and integrating host and media design. Finally, the differences between this study and previous research should be pointed out, and the innovation points of this article should be emphasized. Response: The Innovation points have been added in lines 124-134 2. Relevant reasons should be provided for the selection and concentration settings of seven chemical substances, namely ammonia, pentane, tetrahydrofuran, butanol, phenol, xylene, and benzene, for the application testing of fixed robots. Response: We included a description for the selection of these chemicals in lines 357-360. 3. It is recommended to compare the experimental testing and results with the relevant research results at the end, highlighting the good performance of this study. Response: A general comparison of the results was added in lines 570-577. 4. The title of the test result image of the sensor should be removed from the "Index" to maintain consistency among all image titles. Response: Thank you for this remark. The titles of the figures (5, 6, 7, 8, 9) have been corrected and the “Index” has been removed. 5. It is recommended to appropriately delete the introduction section. Response: Lines 72 -91 have been deleted.

Reviewer 2 Report

Comments and Suggestions for Authors

In this paper, the development and testing of a portable sensor node to detect and monitor the leakage of hazardous chemical gases and vapors during robotics-based operations is presented. The sensor node works with the Internet of Things technology to transfer measured data to database servers and live monitoring and control stations. What’s more, this paper has nice diagrams, clear logic and representative illustrations. It can arouse more interest of readers after a few minor revisions.

1. Page 7, how does the path design of mobile robots affect the response of sensors? And why choose to test through narrow corridors?

2. Page 11, the article mentions some performance limitations of sensors, such as the inability of BME680 to detect acetone or toluene. Did the study mention any strategies or suggestions for improving sensor performance?

3. From the text, it can be seen that the testing distance has an impact on the sensor response. Please explain in detail how distance affects the response of sensors, and whether there are obvious trends or patterns?

4. Given the differences in sensor responses, how may these sensors be practically applied in scenarios where chemical species and leakage rates vary? Are there specific recommendations for selecting sensors based on material characteristics?

5. Some typing errors are found in the manuscript, please double check the writing and the format before a resubmission.

6. Please use a figure to draw the four main nodes of the device.

7. Please explain the advantages of this device compared to other devices of the same type.

8. Please explain the principles used by these three gas sensors for gas detection.

9. There are many and complex test charts for detecting different gases using three types of gas sensors. Please modify and refine them appropriately.

10. Please analyze and explain whether the device has gas selectivity and whether it can identify mixed toxic gases.

11. The repeatability and long-term stability of the device should be given.

12. This work investigated the gas sensors and gas sensing application. Some relative papers may enrich the concepts and background of this work as references: Nano Energy, 2023, 116: 108788.

Comments on the Quality of English Language

Minor editing of English language is required.

Author Response

Dear reviewer, We appreciate your devoted time in reviewing our paper and your valuable comments which enabled improvements in the current version of the manuscript. The authors have carefully considered all comments and tried our best efforts to address every one of them. A point-to-point response is provided below.: 1. Page 7, how does the path design of mobile robots affect the response of sensors? And why choose to test through narrow corridors? Response: The required answers have been added in lines 266-270. 2. Page 11, the article mentions some performance limitations of sensors, such as the inability of BME680 to detect acetone or toluene. Did the study mention any strategies or suggestions for improving sensor performance? Response: The comment has been explained in the added lines in 336-340. 3. From the text, it can be seen that the testing distance has an impact on the sensor response. Please explain in detail how distance affects the response of sensors, and whether there are obvious trends or patterns? Response: The required explanation has been added in lines 348-353. 4. Given the differences in sensor responses, how may these sensors be practically applied in scenarios where chemical species and leakage rates vary? Are there specific recommendations for selecting sensors based on material characteristics? Response: Yes, the responses of the sensors differ depending on the leaking materials, and this was proven in the results of the research paper, and this point was mentioned in 332-340 and 587-596. 5. Some typing errors are found in the manuscript, please double check the writing and the format before a resubmission. Response: The manuscript was extensively revised for typing errors, and a request for proofreading will be submitted if it is accepted. 6. Please use a figure to draw the four main nodes of the device. Response: The required Figure was added to the manuscript (Figure. 1) in lines 146-147 and 158-162. 7. Please explain the advantages of this device compared to other devices of the same type. Response: The required explanation has been added in lines 124-134. 8. Please explain the principles used by these three gas sensors for gas detection. Response: The required explanation has been added in lines 565-569.
9. There are many and complex test charts for detecting different gases using three types of gas sensors. Please modify and refine them appropriately. Response: All the figures have been checked, and the required corrections were made. 10. Please analyze and explain whether the device has gas selectivity and whether it can identify mixed toxic gases. Response: The required explanation has been added in lines 565-567. 11. The repeatability and long-term stability of the device should be given. Response: This point will be processed in future work with a special test environment (without robots). 12. This work investigated the gas sensors and gas sensing application. Some relative papers may enrich the concepts and background of this work as references: Nano Energy, 2023, 116: 108788. Response: Thank you for this remark. The suggested reference was added to the literature in lines 52, and (632-634)

Round 2

Reviewer 1 Report

Comments and Suggestions for Authors

The paper can be accepted.

Reviewer 2 Report

Comments and Suggestions for Authors

The revised version addressed all my concerns. It can be accepted.